# Wearable ECG Device and Machine Learning for Heart Monitoring

**DOI:** 10.3390/s24134201

**Published:** 2024-06-28

**Authors:** Zhadyra Alimbayeva, Chingiz Alimbayev, Kassymbek Ozhikenov, Nurlan Bayanbay, Aiman Ozhikenova

**Affiliations:** 1Department of Robotics and Technical Means of Automation, Satbayev University, Almaty 050013, Kazakhstan; zhadyralimbay@gmail.com (Z.A.); ozhikenovk@gmail.com (K.O.); bayanbay_nur@mail.ru (N.B.); aiman84@mail.ru (A.O.); 2Department of Information Technologies and Library Affairs, Kazakh National Women’s Teacher Training University, Almaty 050000, Kazakhstan; 3Joldasbekov Institute of Mechanics and Engineering, Almaty 050010, Kazakhstan

**Keywords:** wearable ECG device, ECG monitoring system, ECG database, ECG signal, PQRST wave detection, machine learning

## Abstract

With cardiovascular diseases (CVD) remaining a leading cause of mortality, wearable devices for monitoring cardiac activity have gained significant, renewed interest among the medical community. This paper introduces an innovative ECG monitoring system based on a single-lead ECG machine, enhanced using machine learning methods. The system only processes and analyzes ECG data, but it can also be used to predict potential heart disease at an early stage. The wearable device was built on the ADS1298 and a microcontroller STM32L151xD. A server module based on the architecture style of the REST API was designed to facilitate interaction with the web-based segment of the system. The module is responsible for receiving data in real time from the microcontroller and delivering this data to the web-based segment of the module. Algorithms for analyzing ECG signals have been developed, including band filter artifact removal, K-means clustering for signal segmentation, and PQRST analysis. Machine learning methods, such as isolation forests, have been employed for ECG anomaly detection. Moreover, a comparative analysis with various machine learning methods, including logistic regression, random forest, SVM, XGBoost, decision forest, and CNNs, was conducted to predict the incidence of cardiovascular diseases. Convoluted neural networks (CNN) showed an accuracy of 0.926, proving their high effectiveness for ECG data processing.

## 1. Introduction

The human heart is a powerful muscular pump. On a daily basis, the heart undergoes 100,000 contractions and relaxations, pumping 7600 L of blood through the body. Blood circulation removes carbon dioxide, filters out other waste, and supplies oxygen and other nutrients to the body’s cells. Abnormalities in natural blood circulation can cause various heart ailments. According to the World Health Organization (WHO), cardiovascular diseases have long been the leading cause of mortality globally. In the case of Kazakhstan, the mortality rate from cardiovascular diseases (CVD) is two times higher than in European countries. The mortality rate is especially higher for people who experience chronic heart failure (CHF), which develops as a result of coronary heart disease. In the last decade, the rate of cardiovascular disease (CVD) has increased by 1.7 times. Based on collected data, there has been a fourfold increase in hospitalizations for ischemic heart disease compared to 20 years ago. Among Kazakhstan’s 13 million adult residents, 350,000 have been diagnosed with chronic ischemic heart disease [1].

Wearable systems for cardiac disease diagnostics play a key role in healthcare, providing a range of benefits. They assist in detecting and diagnosing heart diseases at an early stage, which facilitates initial treatment and risk reduction. The ongoing monitoring of the cardiovascular system in real time is particularly beneficial for patients with chronic diseases. Wearable systems can be employed to conduct monitoring outside of hospital environments, making them more accessible for various patients, including those who live in remote areas. It reduces the need for frequent doctor visits, saving both time and healthcare costs. Due to constant monitoring and data collection, doctors can effectively prescribe and tailor treatment according to the individual characteristics of patients.

The proposed system includes a wearable device that functions in real time and can promptly warn patients about health conditions. Any wearable ECG device functions by registering human biopotentials, handling noise-resistant signal processing, and diagnosing serious cardiac arrhythmia.

Prior to outlining the main points, the researchers first reviewed various ECG recording wearable devices, noise reduction approaches, and heart disease detection methods that can be employed for creating ongoing health monitoring systems.

Despite the ever-increasing use and commercialization of wearable devices, some constraints hinder the success and usefulness of existing products for health monitoring. Wearable devices are worn as smart watches on the body in a single location (the wrist), therefore limiting their ability to access various biosignals that are detected from different parts of the body [2,3,4].

ECG signals obtained from the left arm’s sensors after signal processing fail to provide recordings of sufficient quality for long-term monitoring. Some cordless devices for ECG recordings and wireless data transmission via Bluetooth and Wi-Fi only record data on heart rate rather than ECG recordings, and none of these devices provide signal quality assurance data that can be directly compared with Holter monitoring data [5,6,7,8]. The design of textile-based multichannel ECG systems that measure ECG signals from several parts of a patient’s torso was described in papers [9,10,11,12,13]. Smart or electronic textiles (e-textiles) are materials that can interact with the environment and users. Designing e-textiles became possible due to agile textile designs that open up the possibility of discreet, versatile, and wearable clothing-based devices. E-textiles also offer the opportunity for the subtle integration of different sensory modes in various parts of the body. Yet, while designing and choosing textile electrodes for long-term ECG monitoring, it is necessary to consider other factors such as methods of integration and preparation, washing and reusability, and the sensory comfort of electrodes on the skin.

ECG processing for noise reduction in wearable ECG devices is crucial for obtaining clear and accurate data. In some studies, several approaches to reducing the noise in wearable ECG systems have been suggested. For example, digital filters [14,15] are employed to remove unwanted noise components, including high-frequency interference (electrical interference from household appliances) and low-frequency artifacts such as a patient’s movement or respiratory activity. The study by [16] offered methods for eliminating basic lines to remove slowly changing signals caused by electrodes, signal issues, and other sources of artifacts. Moreover, recent machine learning methods were used for detecting and classifying noise components of signals with their subsequent removal or correction [17]. However, most current studies are experiment-based and assess the efficiency of individual filters. Currently, there is a lack of studies that thoroughly examine the various types of filters for noise suppression and the extraction of necessary signals.

Using artificial intelligence (AI) methods to predict cardiovascular diseases has been actively studied in contemporary research. The studies of [18,19,20] used a support vector method (SVM) and other machine learning classifiers to detect signs of cardiovascular diseases from ECG data obtained from patients with heart disease who wore wearable devices. The results show that the SVM classifier is highly effective compared with other algorithms. An automatic classification system for detecting cardiac arrhythmia was developed and studied [21]. The researchers [22] presented an automatic ECG arrhythmia classifier based on a machine learning method called echo state network (ESN). This classifier needs only one ECG lead and demonstrates effective performance in two ECG databases. The paper [23] proposed a simple and cheap algorithm for processing and analyzing ECG signals by using linear regression for signal segmentation and detecting important components. The use of deep learning, including models that search for QRS and T-waved vectors, was studied [24,25].

These models have demonstrated high accuracy for large public datasets of clinical ECG recordings. In [26], algorithms were developed to calculate the R-R interval with high accuracy using data from the MIT-BIH database.

The researchers [27] introduced a wearable ECG monitor integrated with wireless sensors for obtaining ECG signals and classifying them with machine learning methods. However, the accuracy of this system still needs improvement. A new artificial neuron network was offered by [28] for the reliable identification of atrial fibrillation in the ECG signals. Moreover, a basic model with a recurrent neural network (RNN) and a lightweight model with a cast RNN were used [29] to accelerate the prediction time. The paper [30] proposed a method based on a neural network for the automatic identification of connections between the conditions of elderly patients and various factions calculated from ECG and EEG signals. Although the above-mentioned studies prove the potential of implementing AI in the management of cardiovascular diseases, further studies are necessary to verify the effectiveness of these methods in real-time scenarios.

This review demonstrates the necessity of developing a wearable system capable of providing consistent ECG signal processing and detecting serious cardiac arrhythmias in real time, during the patient’s unrestricted activities. At present, there are no wearable ECG monitoring systems that provide continuous heart monitoring for patients during free activity. Existing market devices, such as smartwatches, patches, and textile-based wearables, have limitations, particularly their inability to detect all the waves and segments of the cardiac cycle. Most of these devices only detect the R-peak, which limits their diagnostic capabilities to heart rate (HR)-related conditions. Devices that can capture a full ECG signal (e.g., Holter monitors) lack the capability for immediate medical intervention since analysis results are available only 24–72 h after recording. Therefore, current methods that employ artificial intelligence for the detection of heart abnormalities focus on using signals from devices that only measure HR or on pre-recorded ECG data.

In this work, we propose a wearable ECG monitoring system composed of three levels (ECG device, server, and physician’s workstation) that provides a comprehensive ECG signal (including all waves and segments) and detects heart abnormalities during the patient’s free activity. This is achieved through the complex filtering and preprocessing of the cardiac signal, as well as real-time machine learning methods.

The novelty of this research lies in the development of an integrated ECG monitoring system designed for use during free activity, enabling real-time heart monitoring. This system implements all stages of ECG signal processing and analysis, from filtering to the detection of cardiac abnormalities, in practical applications (shown in Figure 1). This article describes all stages in detail, including hardware development, the pre-processing of ECG signals, methods for detecting severe cardiac arrhythmias, and machine learning methods for predicting heart diseases based on data obtained from wearable ECG devices.

The cardio diagnostics system includes a wearable ECG device, software complex (website), and server, and performs a number of important functions. This system is constantly recording electrical heart signals, pre-processing signals for noise suppression and providing accurate data, highlighting informative parameters for further detailed analysis, forming diagnostic features based on the extracted parameters, conducting a detailed assessment of the heart’s condition, and utilizing processed data to detect possible dysfunctions or alternations in cardiac system functioning.

## 2. Materials and Methods

### 2.1. Hardware Development of Wearable ECG Devices

The ECG data necessary for the study were obtained from a wearable ECG device built by the authors, as shown in Figure 2.

A wearable ECG device is one of the key elements of a cardiology diagnostic system. It facilitates the analog-to-digital conversion and transmission of ECG signals through radio channels to a smartphone.

Figure 3 illustrates the detailed functional flow chart of the proposed ECG device.

A wearable ECG device consists of several main components, among which the most important are microcontrollers.

ADS1298, connected to the microcontroller STM32L151xD through a serial data transmission channel, was used to collect ECG data.

Ensuring an independent power supply for ADS1298 to minimize interference is a main stage of the process. The non-volatile, high-speed memory block, MRAM, was employed to record operative data, while for long-term data storage, an external microCD Flash with a 32 GB capacity was used during data collection. In addition, the ECG device was equipped with a MEMS accelerometer to determine the patient’s heart-attack-induced falls and immobility. Power and battery charge are supplied by an external source of power using a charging device, which tracks the actual battery charge. There is also a power button controller and a residual battery charge detector. A GPRS communication module was used to transfer data to both the medical information system and the ambulance service, while a Bluetooth module was employed to send through a mobile phone. Moreover, a GNSS module was used to identify a patient’s location. The visual control system includes three-color LEDs that display various operating modes and single-color LEDs for signaling about the connection to the Internet and data exchange on the server.

Figure 3 also shows the order of attachment of the ECG electrodes to the patient by cable color, with the specified electrode locations on the torso:

Electrode 1—Red cable—Second intercostal space at the right sternum;

Electrode 2—Yellow cable—Fourth intercostal space at the right sternum;

Electrode 3—Green cable—Fourth intercostal space at the left sternum;

Electrode 4—Black cable—Fifth intercostal space at the right sternum.

### 2.2. Software Complex

The following tasks were carried out during the development of software components:-Algorithms for extracting data from a microcontroller that records ECG data from patients were developed.-A web component of the system was built that medical workers could interact with.-A server module, responsible for the recording of data from a microcontroller in real time and its transmission to a web component, was built.-Methods of processing initial data on the server were studied and implemented, along with mechanisms for transmitting processed data to the web component.

A Java Script framework with an open-source Vue.js was chosen to create the user interface for the client module of the cardiac diagnostics system. Vue.js has convenient integration with other Java Script libraries and can be employed to build a single-page application. This framework enables a user interface to be created based on the Model-View-ViewModel architecture template. Vue.js specializes in demonstrating the level of easy integration with other projects and libraries. Websites built with Vue.js can be efficiently uploaded due to their small size (17 Kb).

The free framework Django, written in Python, was chosen for the software part of the service (the backend). Django follows the design pattern MVC (Model-View-Controller) and enables web applications to be created from separate plug-in applications. This framework adheres to the principle of DRY (Do not repeat yourself), which prevents codes from being repeated. One of the features of Django is a URL handler configuration that uses regular expressions. ORM (Object-Relational Mapping) can be used to describe data models in Python and automatically create data base schemes that function in the Django data base. ECG data were retrieved from a server using a REST API (Representational State Transfer) architecture style. All retrieved data was stored in the object-relational database management system PostgreSQL, a flexible and reliable database management system that supports user objects and complex data structures.

Each user has their own unique data, which is sent to the client module in Vue.js with the REST API. The web service collects data every second and visualizes it as a graph that moves in sync with the user’s heart rate. After entering a username and password, the website sends a request to the server, and the requested data appears on the screen.

A JSON Web Token, a token for authentication and user authorization, was used in Python to check user access rights to certain resources. JWT enables the creation of token access to JSON, which confirms the certain privileges of a user. The tokens can be employed to transfer identification data between the client and server.

### 2.3. ECG Signal Filter

Three-stage noise removal techniques were implemented in the proposed ECG cardio monitoring system, which includes the following flowing processes:

At the first stage, filtering is applied for the analog-to-digital conversion of module 1298:-The RC filter at the entrance, with a bandwidth of 3 MHz, functions as a filter for electromagnetic interference in all channels.-Low-pass filters enable frequencies below a specified boundary frequency (cut-off frequency) to pass, thus reducing or eliminating high-frequency noise and interference that distorts signals.-High-pass filters enable frequencies above a specified boundary frequency (cut-off frequency) to pass, thus reducing or eliminating low-frequency noises and constant components of signals, such as breath artifacts or electrical interference.-To ensure the integrity of the sampled ECG signals, anti-aliasing filters were applied before sampling. These filters are designed to prevent aliasing, which can cause distortion and inaccuracies in the digitized signal.-Network filters remove electrical interference from alternating current sources, such as a standard network with a frequency of 50 or 60 MHz, which can create interference with the ECG signals.

At the second stage, the Median filter was applied to the hardware on the controller. This method includes defining the filter window, implementing median filtering for each signal count in the window, processing boundary conditions, and repeating the process to improve the filter efficiency. The median filter is effective in removing impulsive and other noises, keeping the main features of signals.

At the third stage, as a software application using digital signal processing based on filtering algorithms, a bandpass filter is implemented. The bandpass filter is adjusted to a certain frequency range containing cardiac activity and blocks frequencies beyond the range. It eliminates not only frequencies that are equal to the electrical activity of the heart, but also blocks interference from other sources, such as electromagnetic interference or network noise, as well as low-frequency artifacts, such as body movement or electric motor activity.

A bandpass filter involves a system that allows a frequency signal (*f*) in a certain range (*fc*) to pass and suppresses frequencies outside this range. Mathematically, it can be described through the transmit function *H*(*f*), which is identified as:(1)$$H(f)=\frac{1}{\sqrt{1+{\left(\frac{f}{f_{c}}\right)}^{2n}}},$$

The function *H(f)* decreases as frequency increases *f*, reaching half of its maximum value at the cut-off frequency *fc.* The parameter n identifies the steepness of the frequency response of the filter; the more n, the sharper the transition from passing to suppressing the signal around the cut-off frequency.

### 2.4. ECG Signal Clustering

In the process of analyzing electrocardiogram (ECG) data, situations may arise where the signals have inverted polarity due to technical aspects of the recording equipment or errors during the recording process. Inverted polarity can lead to an erroneous interpretation of cardiac activity, which is a critical issue in clinical diagnostics.

To address this issue, a K-means algorithm was employed to classify ECG signals based on the similarity of their amplitude characteristics and waveforms. The clustering process began with determining the number of clusters and selecting initial points for the centroids. Each data point was then assigned to the cluster with the closest centroid based on the Euclidean distance. The centroids were recalculated until convergence was achieved, indicating the stabilization of the clusters.

After clustering, the polarity of the signals within each cluster was analyzed. If the polarity of a cluster was determined to be abnormal (inverted), a polarity correction was performed:(2)$${ECG}_{corrected}\left[i\right]=-1\times ECG\left[i\right]$$
where *ECG*[*i*] is the amplitude of the *i*-th signal in the cluster with abnormal polarity. This transformation restores the expected normal orientation of the signal, improving diagnostic accuracy and enhancing the quality of the data for subsequent analyses. This approach is crucial for ensuring accuracy in identifying *ECG* elements, such as R peaks, and assessing cardiac intervals, which are critical for diagnosing cardiac pathologies.

### 2.5. PQRST Wave Detection

The precise detection of ECG signal features, such as the R peak, QRS complex, and other components of waveforms, is crucial for detecting cardiac abnormalities. Traditional ECG signal analysis means manual interpretation, which takes plenty of time and is prone to human error. Thus, there is a growing interest in designing automated methods for improving the preciseness and effectiveness of ECG signal analysis.

The process of detecting peaks P, Q, R, S, and T in ECG signals involves initially identifying the R peak. This is followed by detecting other significant peaks by analyzing changes in amplitude and temporary intervals between these peaks and other points on the ECG signal.

The Q point is identified in the interval before the R peak. A time interval in the form of a Q interval is used, which is typically approximately 0.04 s (or 14.4 samples at a sampling rate of 360 Hz). Thus, to search for the Q point, the algorithm analyzes the signal in the interval [R−Qinterval,R], where R is the index of the peak R.

The S point is identified after the R peak. Sinterval, with 0.08 s (or 28.8 counts at the same frequency sampling), is used as the S point. The S point search interval is [R, R+Sinterval].

The T point represents elevation, following the S wave, and illustrates a period of repolarization in the ventricles of the heart. The interval after the S point is used for searching for this. The T interval usually starts immediately after the S point and can last until 0.2 s or 72 counts. Detecting T points involves searching for the maximum value of amplitude within this interval.

The P wave illustrates a period of atrial depolarization and precedes the QRS complex. The interval before the Q point is used to detect it. The P interval can vary but is often selected from 0.12 to 0.20 s before the Q point. Searching for the P point involves identifying the maximum amplitude within the interval.

This process introduces the basic methods for detecting important points in the ECG signal. It is worth mentioning that detecting preciseness depends on signal quality, the presence of noise and artifacts, and certain parameters of the algorithm in signal processing.

### 2.6. Anomaly Detection in ECG Signals

In this study, we adopted the Isolation Forest, a method for detecting abnormalities, to collect data consisting of key point labels in an ECG (waves Q, R, S, T, and P). We aimed to detect potential abnormalities in cardiac rhythms, which could indicate the presence of abnormalities or other irregular cardiac conditions. Isolation Forest is a machine learning algorithm based on ensemble decision trees that is used to detect abnormalities by isolating individual data points.

The algorithm was set using 100 estimators (‘n_estimators = 100’), where parameter ‘contamination’ was set as ‘auto’ for automatically detecting the ratio of abnormalities in the dataset. The arbitrary selection of subsamples and divisions within trees was controlled with the parameter ‘random_state = 42’, ensuring the consistency of the results.

Once the model was trained on an ECG dataset, the abnormal condition of cardiac rhythm was predicted. Algorithm predictions categorized the dataset into normal (designated as “1”) and anomalous means (designated as “−1”). These results were integrated back into the initial dataset, enhancing it with abnormal labels for each ECG record.

### 2.7. Predicting Heart Disease

The research aims to analyze relationships between various factors (age, sex, chest pain, blood pressure, cholesterol, alcohol, diabetes, ECG change, smoking) and a patient’s health condition.

To carry out the study, 132 participants were chosen from the cardiology center of the Central Clinical Hospital of Almaty, Kazakhstan, and their data were recorded. The study included both men (98 participants) and women (34 participants). Their ages varied from 28 to 68. The participants’ average blood pressure was 129.93 mmHg, with a minimum value of 94 mmHg and a maximum value of 200 mmHg; their cholesterol level was within the minimum level at 126 mg/dL and the maximum at 341 mg/dL. In total, 43.18% of participants reported using alcohol, 31.06% of participants had diabetes, and 55.30% of them were smokers, while changes in ECG data were observed in 37.12% of participants. Data were collected using conventional cardiac methods of assessment, including measuring blood pressure and cholesterol level, as well as conducting an ECG study using a wearable system built by the authors. This study aimed to analyze the relationship between lifestyle, physiological parameters, and the risk of developing cardiac diseases. All participants were requested to abstain from alcohol, coffee, or tea for 12 h prior to the experiment. Before the experiment, each subject underwent a checkup, where we measured their blood pressure and consulted their medical history. Furthermore, during the experiment, we employed a wearable ECG device for recording the ECG signals. The results were verified by an experienced cardiologist, and he made notes to collect data for further research.

The participants’ data were divided into 10 parameters. Information about dataset attributes is illustrated in Table 1.

The aim of this study was to assess 10 parameters: the presence of cardiovascular disease (CVD) in patients and the first 9 parameters shown in Table 1.

During the study, various machine learning methods were applied, including logical regression, decision trees, random trees, support vector machines (SVM), XGBoost, and convolutional neuron networks (CNN), followed by a comparative analysis and investigation of the efficiency of each method.

## 3. Results

### 3.1. ECG Signal Filter

As mentioned in Section 2.3, the process of filtering ECG signals with the aim of suppressing noise was implemented concurrently in three stages; two of them were executed at the hardware level. Particularly, the analog-to-digital conversion was carried out by applying an RC filter, high- and low-frequency filters, digital thinning filters, and network filters. In the microcontroller, the median filter was involved in removing impulse interference. As a result, the ECG signal was obtained, as illustrated in Figure 4. Artifacts were detected that hindered subsequent signal analysis.

Signal filtering is crucial for preparing ECG signals for further analysis, as it improves visualization, which is important for observing components such as the R peak, and facilitates automatic detection. A bandpass filter was used to improve the precision and reliability of diagnostic outcomes.

The following parameters were set for filtering the ECG signals:

A low-cut range of 0.5 Hz, which removes low-frequency oscillations that are not related to cardiac activity.

A high cut range of 100.0 Hz, which facilitates the suppression of high-frequency noises while maintaining the important components of ECG signals.

The sampling frequency (fs) of the ECG signal is 360 Hz, a typical value for medical ECG data, ensuring the overall quality of signal recovery.

Segments after ECG signal filtering are illustrated in Figure 5, which shows the effectiveness of the implemented ECG data processing.

Pre-filtering the signals enabled the removal of some artifacts and highlighted the main artifacts for analyzing signal functions.

### 3.2. Clustering ECG Signals

Clustering enables signal points to be classified by amplitude, facilitating visualization and data interpretation. This algorithm requires the number of clusters to be specified (in our case, they are 5). It scales well to large numbers of samples and is used in a wide range of ECG signal processing applications.

For the analysis of ECG data, we employed the K-means method with five clusters. This choice was based on a preliminary analysis of signal characteristics and experiments conducted with varying numbers of clusters. We found that using five clusters allows for the most adequate division of data, highlighting significant signal features. Such an approach ensures an accurate representation of key ECG components and enhances subsequent data interpretation.

Figure 6 shows a graph containing data from the electrocardiogram and the processing results from the clustering algorithm. The *X* axis is the duration of cardiac activity, while the *Y* axis represents the amplitude of the electrical signal.

The amplitudes were inverted to improve visualization and analyze individual clusters of signals (Figure 7). This step enabled the polarity of signals to be corrected and improved the detection of key points in the ECG.

As seen in Table 2, there are 5 clusters, and in Figure 6 and Figure 7, they are depicted in different colors.

The algorithm grouped the data points of ECG signals detected into features of the presented signals based on their similarities. Clusterization highlights the differences in heart rate characteristics identified in ECG signals.

As a result of clustering using the K-means algorithm, an actual visualized ECG signal was obtained, as illustrated in Figure 8.

The data obtained through the filtering and clustering processes were processed to determine the PQRST wave in the ECG signal.

### 3.3. PQRST Wave Detection

The process of identifying points P, Q, R, S, and T on an ECG involves detecting R peaks and identifying other points using amplitude analysis and temporary intervals within them. The Q point was identified prior to the R peak by applying the temporary interval, Qinterval. The S point was identified after the R-peak by applying the interval, Sinterval. The T point was identified after the S waves. The P point was detected prior to the QRS complex.

The results of detecting PQRST waves can be seen in Figure 9.

As seen in Figure 9, the peaks of points P, Q, R, S, and T are depicted in certain colors: yellow (P point); green (Q point); and red (R point). Peaks designated by blue are the S point, and purple peaks are the T point.

The forms of PQRST waves are almost the same as described in Section 2.5, meaning that the wearable system was effective enough to detect waves in an ECG signal.

### 3.4. Anomaly Detection in ECG Signals

The results obtained from this study reveal the importance of a complex approach to analyzing ECG signals, including methods of digital signal processing and clustering. In Table 3, we kept the identified labels with data for further analysis, utilizing machine learning methods. Table 3 presents a sample of the labeled data, including indices of various key points (Q, R, S, T, and P) in the ECG data.

These data are used to analyze the characteristics of the ECG signal and classify different phases of the cardiac cycle.

The analysis of the results demonstrated that a certain number of records were classified as “anomalous”, while others were regarded as “normal”. A qualitative assessment revealed a distribution between anomalous and normal states (Table 4) that indicates the potential presence of irregular patterns in the dataset retrieved from the ECG signals. Table 4 contains readings of the detected anomalous and normal data.

These data help assess the effectiveness of algorithms in classifying normal and anomalous ECG records, which is important for improving diagnostic accuracy and predicting patients’ health conditions.

We applied the Isolation Forest method to classify 56 cardiac rhythm records without prior anomaly labels. As a result, the model correctly classified 20 out of 21 actual anomalous records, which accounts for 97% of the total anomalous records. This yields the following metrics: accuracy of 97%, true positive rate of 95%, and false positive rate of 5%. The remaining 36 records were classified as normal (Figure 10).

To configure the Isolation Forest model, we selected parameters that balanced accuracy, performance, and reproducibility of results. The number of trees in the ensemble was set at a level that ensured stable predictions. The proportion of data considered anomalous was adjusted so that the model could automatically assess this fraction based on the data. Additionally, an initial value was set for the random number generator to ensure the reproducibility of results in the future. These parameters were chosen based on common practices and experiments to achieve optimal model performance in ECG data analysis.

Further analysis demonstrated that the average values of Q, R, S, T, and P waves in the index of anomalous records differ from the average values of normal records. In particular, anomalous records manifested the early presence of all waves in cardiac rhythm compared with normal records. It can indicate certain changes in cardiac rhythm associated with abnormalities.

In Figure 11, the y-axis represents the amplitude of the ECG signal at each time index. This index facilitates the visualization of amplitude changes in the signal over time.

Rectangular graphs may appear similar for different points, such as R peaks and points P, Q, S, and T, due to the normalization process applied before clustering. During normalization, each segment of the ECG signal is scaled to a comparable range, enabling sequential clustering based on the waveform rather than the signal amplitude.

Although the raw amplitudes of R peaks are typically much higher than those of points P, Q, S, and T, normalization is necessary to prevent amplitude bias during clustering. This approach allows segments with similar shapes to be grouped, which is crucial for the accurate identification and analysis of various components of the ECG signals.

These results are important for identifying potential abnormalities in ECG data, which can offer a justification for further research into cardiovascular diseases. Additional analysis, including comparison with medical data and initial diagnoses, is required to confirm the clinical importance of detected abnormalities.

### 3.5. Predicting Heart Disease

Our data set includes information about the group of patients, explained thoroughly about their demographic characteristics, lifestyles, and indicators that show the health status of the subject. This creates a basis for studying the relationship between “decommunization” and lifestyle choice and the consequences of this for health. We used unprocessed data from 132 patients, which we collected as a result of using our ECG device. The data fraction is given in Table 5.

Prior to conducting a more detailed analysis of our data, a statistical analysis was conducted to evaluate the direction of parameter interrelations. Correlations between each pair of features were analyzed and identified.

Age—patient’s age.Sex—gender of the patient.Chest pain—is there pain in the patient’s heart or not?Blood pressure—the patient’s blood pressure during the examination.Cholesterol—cholesterol levels.Alcohol—does the patient drink alcohol or not?Diabetes—does the patient have diabetes?ECG Change—a change in the patient’s ECG.Smoking—cigarette use?Condition—sick or not?

The coefficient of correlation between variables is illustrated in Figure 12. Each table cell represents a correlation between two parameters in our dataset. Correlational analysis shows the significance of the relationship between some features and the health conditions of patients.

Figure 12 shows the correlation coefficient between each pair of features. However, as noted, correlation coefficients below 0.6 may not provide convincing evidence of significant relationships.

For a more comprehensive analysis, we included boxplots, which visualize the distribution of data and help identify potential dependencies between variables and states. These graphs clearly demonstrate how various factors correlate with patient conditions, enhancing our understanding of the data and confirming the findings of the correlation analysis.

Figure 13 displays boxplots for the following variables: age, gender, chest pain, blood pressure, cholesterol, alcohol consumption, diabetes, ECG changes, and smoking. These graphs visually assess the distribution of each variable’s values depending on the condition (0 or 1). For instance, it can be observed that age, cholesterol level, and blood pressure exhibit a wider range of values among patients with condition 1. Similarly, ECG changes and diabetes are more prevalent among patients with condition 1.

The use of boxplots allows for a clearer visualization of the data and complements the results of the correlation and regression analysis, confirming the significance of various factors for the patients’ condition.

The researchers applied statistical and machine learning methods, including logical regression, decision trees, random forests, support vector machines (SVM), and convolutional neural networks (CNN), for analysis. The efficiency of the model was assessed using an accuracy label and the percentage of accurate responses from the model in relation to all predictions, enabling the overall effectiveness of the model to be evaluated.

Multiple random divisions of preprocessed data into training and evaluation subsets were applied to evaluate the machine learning method. Furthermore, features were scaled to improve the training process and accuracy of the model. Scaling gives each feature a standardized form with an average value of 0 and a standard deviation of 1.

The graph shows the accuracy and F1 score values for each model.

The graph was illustrated to compare the accuracy and F1 score values for each model and analyze the retrieved data (Figure 14).

The results of training and assessing the performance of convolutional neuron networks (CNN) for binary classification tasks demonstrate the dynamics of model training and its ability to be classified based on the given data.

Applying various models of machine learning indicated their various degrees of efficiency in predicting patients’ health conditions. Convolutional neuron networks demonstrated remarkable performance, reaching an accuracy of 0.926, which indicates their potential for processing complex data patterns.

This research highlights the importance of a comprehensive analysis of medical data in predicting health conditions. The efficacy of integrating traditional machine learning with deep learning techniques opens new perspectives on designing prediction models in medicine.

## 4. Discussion and Conclusions

As research has gradually progressed, it has become clear that concerns related to the cardiovascular system can be predicted by analyzing deviations, detected in ECG data.

This study presents a wearable device for ECG monitoring and an automatic system for predicting diseases based on machine learning methods that enable irregularities in cardiac activity to be recognized. As our findings indicate, this system functions effectively. Our proposed electrocardiography monitoring system is the first of its kind, covering all steps of the monitoring process. These steps include collecting ECG signals, comprehensive signal filtration for suppressing noises, the automatic recognition of all segments and waves in ECG, the illustration of signals in real time, ECG data transmission, and diagnostics based on ECG analysis.

An ECG signal-registering device was constructed based on ADS1298 and the microcontroller STM32L151xD. A Java Script framework with open-source Vue.js was chosen to create the user interface for the client module of the cardiac diagnostics system, while the Django web framework, written in Python, was chosen for the hardware and software modules of the service. REST AP, an architectural style, was used to retrieve ECG data. RC filters, low-frequency filters, high-frequency filters, digital thinning filters, median filters, and bandpass filters were employed to suppress noise from ECG data. Machine learning methods were used for several purposes, including a K-means clustering algorithm to retrieve features and an Isolation Forest algorithm for detecting abnormalities in ECG.

To predict cardiac diseases, there was a comparative analysis of various machine learning methods, including logical regression, decision trees, random trees, support vector machines (SVM), XGBoost, and convolutional neuron networks (CNN), and in subsequent steps, their efficiency was studied. Convolutional neuron networks demonstrated remarkable performance, reaching an accuracy of 0.926, which indicates their potential for processing complex data. Data were collected from 132 participants, 98 men and 34 women, to conduct the experiment.

This system addresses multiple common issues associated with wearable devices available on the market today:There is a need for wearable devices with a large number of electrodes (8–12 channels) to obtain ECG signals containing all peaks, intervals, segments, and complexes.There is an issue with ECG signal filtration in wearable devices, where not all types of noises and artifacts are removed, thus affecting the quality and accuracy of signals and thus hindering the diagnosis process.In current wearable devices, there is a lack of algorithms for automatically identifying deviations in ECG signals, where usually only diseases associated with intervals between RR heart rate are detected.One’s own data cannot be employed to predict cardiovascular diseases using machine learning methods. Current systems for the automated detection of cardiac diseases based on machine learning methods tend to use ECG datasets, such as PhysioNet, that confine their applicability to certain patients or situations.

Despite obtaining significant results, the current study is limited by the fact that the developed algorithms aim to detect pathologies in cardiac activity. However, the algorithms fail to distinguish between different types of heart disease. This means that the system is not fully suitable for implementation in clinical practice. Currently, in collaboration with the Cardiology Center in Almaty City, Kazakhstan, a study is being conducted on developing algorithms to detect myocardial infarction, and the proposed wearable system for ECG monitoring will be integrated. This decision is to improve the functionality of the system and widen its clinical use.

The developed system offers significant potential to enhance the monitoring of the cardiovascular system and to rapidly detect diseases that can lead to a notable reduction in the risk of cardiovascular diseases in patients.

## Figures and Tables

**Figure 1 sensors-24-04201-f001:**
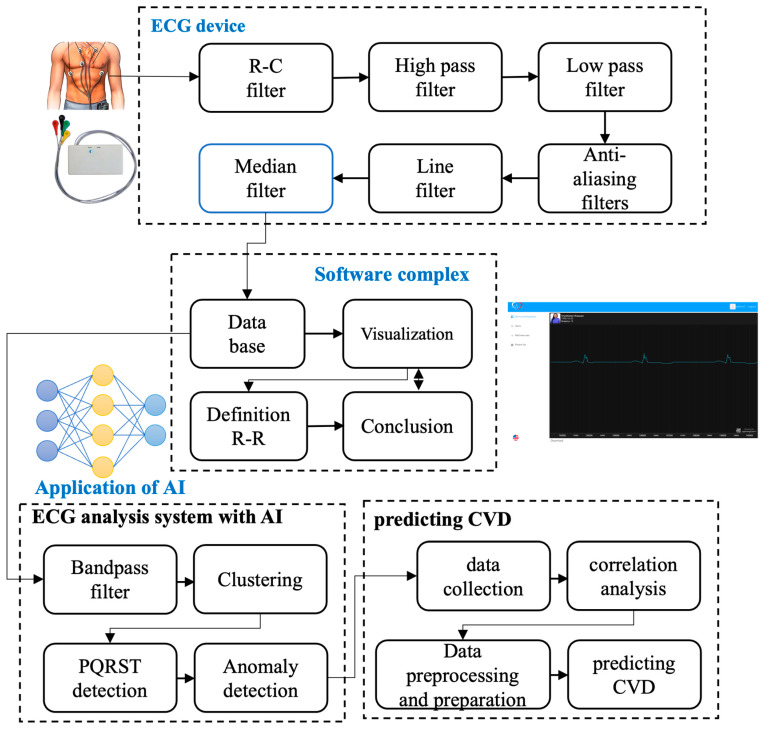
Structure of a wearable system for cardio diagnostics.

**Figure 2 sensors-24-04201-f002:**
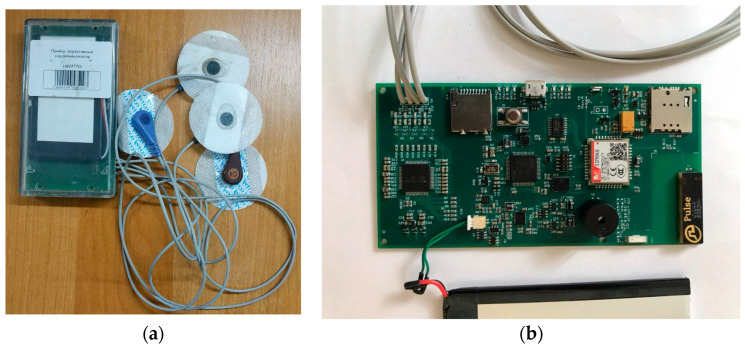
A prototype of a wearable ECG device developed at Satbayev University: (**a**)—external view; (**b**)—board.

**Figure 3 sensors-24-04201-f003:**
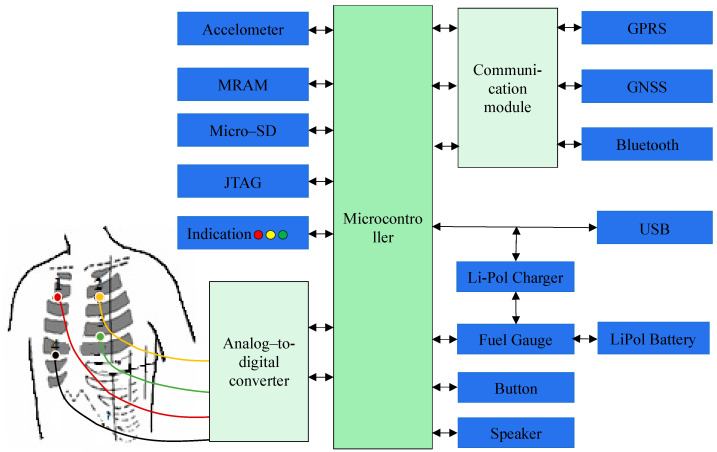
Functional block diagram of an ECG device.

**Figure 4 sensors-24-04201-f004:**
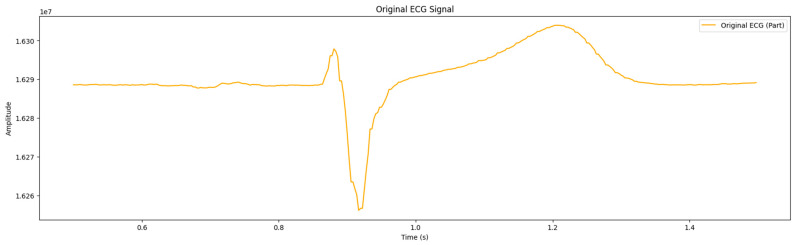
Retrieved signals from a wearable ECG device.

**Figure 5 sensors-24-04201-f005:**
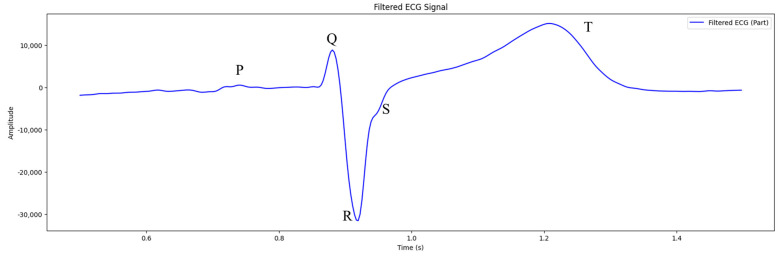
ECG signals after the software implementation of a bandpass filter.

**Figure 6 sensors-24-04201-f006:**
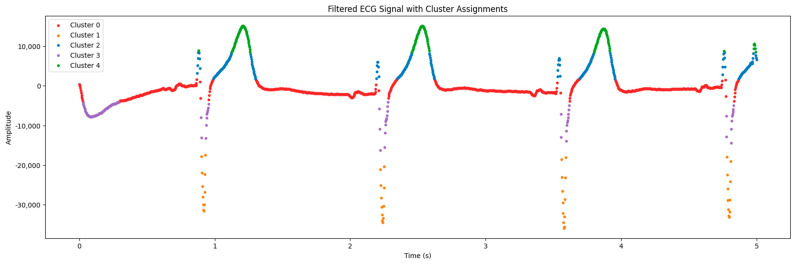
Clustering signals by their features.

**Figure 7 sensors-24-04201-f007:**
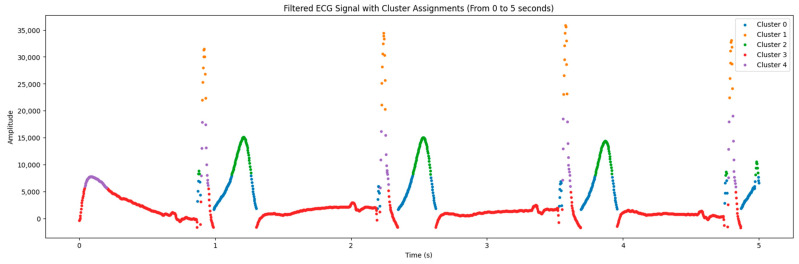
The results for the correct polarity of the signal.

**Figure 8 sensors-24-04201-f008:**
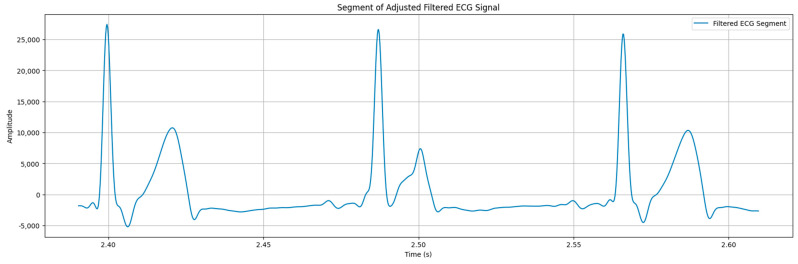
An actual visualized ECG signal.

**Figure 9 sensors-24-04201-f009:**
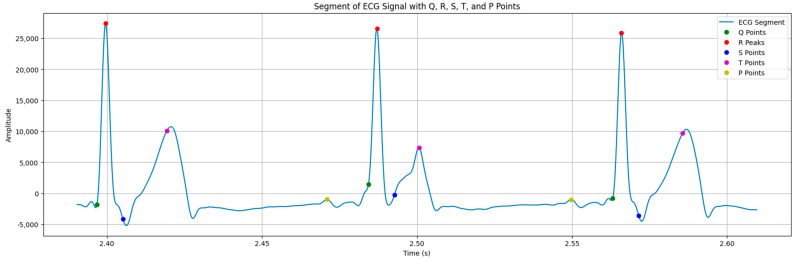
PQRST wave detection.

**Figure 10 sensors-24-04201-f010:**
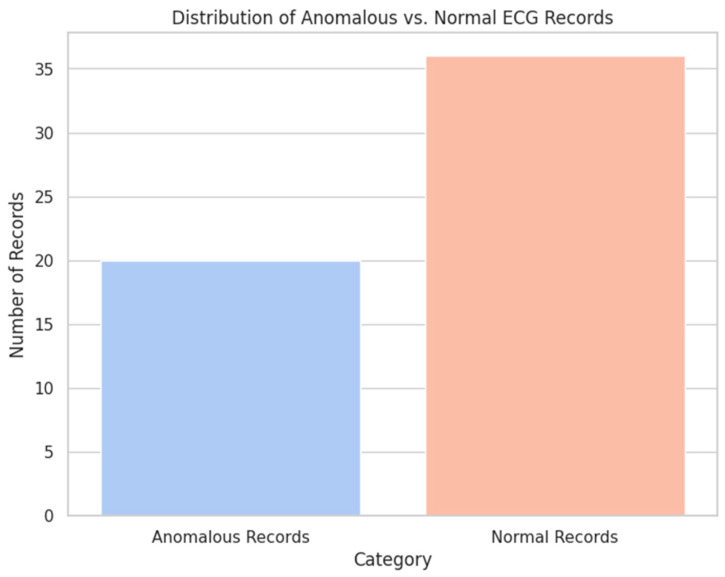
Results of ECG signal analysis revealed by the Isolation Forest model.

**Figure 11 sensors-24-04201-f011:**
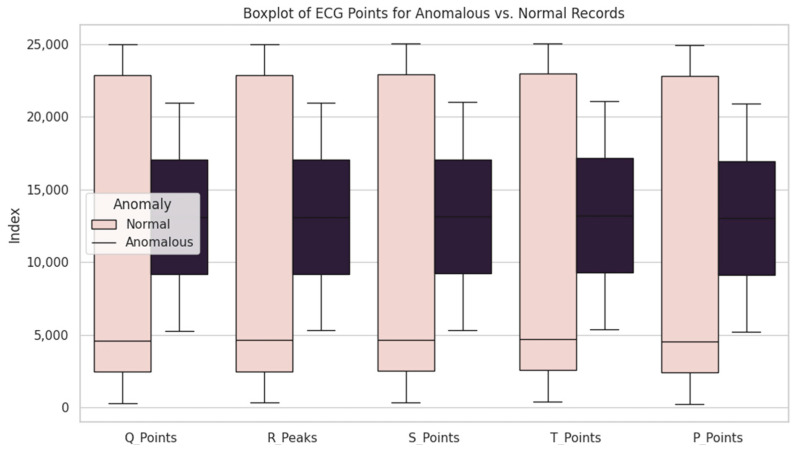
Illustration of variations between anomalous and normal records.

**Figure 12 sensors-24-04201-f012:**
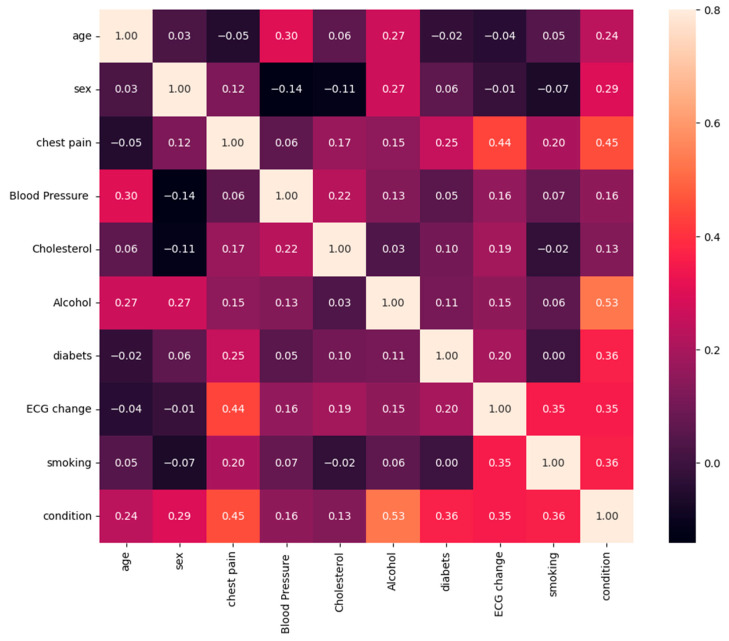
Correlation between each pair of features.

**Figure 13 sensors-24-04201-f013:**
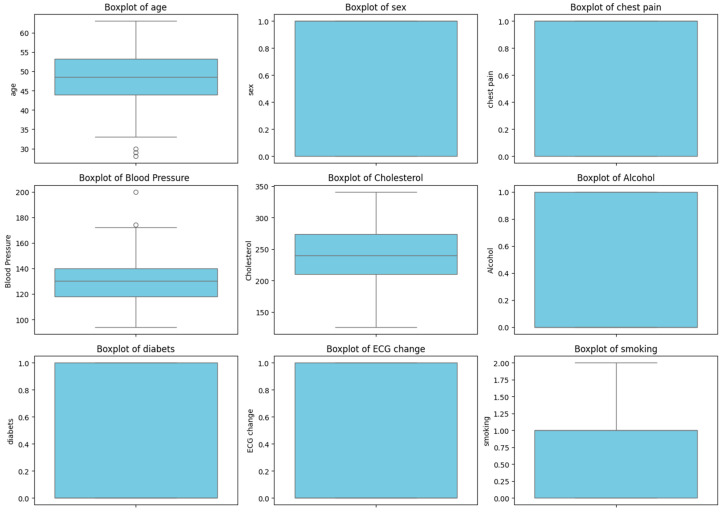
Distribution of features and their effects on health conditions.

**Figure 14 sensors-24-04201-f014:**
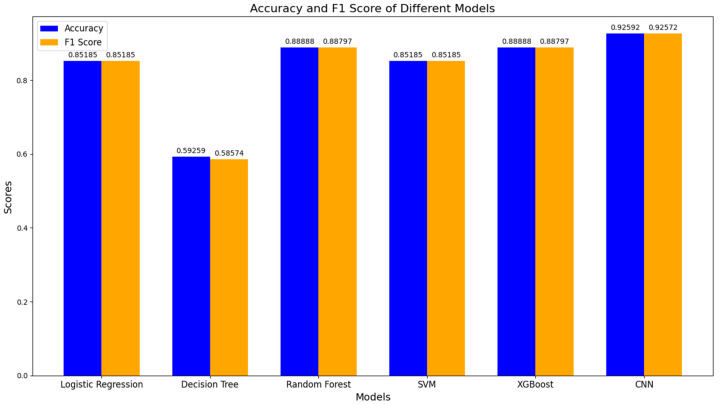
Performance of various machine learning models.

**Table 1 sensors-24-04201-t001:** Description of data attributes.

No.	Characteristic	Category	Number of Patients	Percentage
1	Sex	Male	46	67.65%
		Female	22	32.35%
2	Age (Range)	25–65	68	100%
3	Blood Pressure (Range)	110–178	68	100%
4	Diabetes	Yes	9	13.24%
		No	59	86.76%
5	Heart Attack	Yes	8	11.76%
		No	60	88.24%
6	ECG at Rest	Normal	45	66.18%
		Abnormal	23	33.82%
7	Vessels (0–3)	0	45	66.18%
		1–3	23	33.82%
8	Max (Range)	201–304	68	100%
9	Condition	Healthy	24	35.29%
		Sick	44	64.71%

**Table 2 sensors-24-04201-t002:** Distribution of data points across clusters.

Cluster	Number of Points
0	3420
1	546
2	2057
3	18,364
4	687

**Table 3 sensors-24-04201-t003:** An excerpt of data with labels.

Index	Q_Points	R_Peaks	S_Points	T_Points	P_Points
0	316	330	358	429	248
1	791	805	833	904	724
2	1273	1287	1315	1386	1207
3	1712	1726	1754	1794	1645
4	2107	2121	2149	2220	2040
5	2582	2596	2624	2695	2515
6	3064	3078	3106	3177	2997
7	3503	3517	3545	3584	3436
8	3898	3912	3940	4011	3831
9	4373	4387	4415	4486	4306

**Table 4 sensors-24-04201-t004:** The readings of detected anomalous and normal data.

Index	Anomalous Means	Normal Means
Q_Points	11,858.55	13,112.25
R_Peaks	11,872.55	13,126.25
S_Points	11,900.55	13,154.25
T_Points	11,963.55	13,217.444444444445
P_Points	11,792.4	13,045.25
Anomaly	−1.0	1.0

**Table 5 sensors-24-04201-t005:** Fraction of data collected for the study.

Age	Sex	Chest Pain	Blood Pressure	Cholesterol	Alcohol	Diabetes	ECG Change	Smoking	Condition
35	1	1	120	198	1	0	1	1	1
35	1	1	126	282	1	1	1	0	1
42	1	1	136	315	1	0	1	1	1
48	1	1	124	274	0	1	0	1	1
44	1	1	120	169	1	1	1	1	1
28	1	1	104	208	0	0	1	1	0
45	0	1	138	236	0	0	1	1	0
47	1	1	112	204	0	0	0	0	0
30	0	1	138	243	0	0	1	1	0
46	1	1	140	311	0	0	1	1	1
53	1	1	140	203	1	1	1	1	1

## Data Availability

The datasets generated and analyzed during the current study are available as follows: 1. Signal retrieved from a wearable ECG device: The dataset corresponding to Figure 4 is available on figshare at https://doi.org/10.6084/m9.figshare.26044390.v1 (accessed on 15 June 2024). 2. Identified labels: The dataset extending Table 3 is available on figshare at https://doi.org/10.6084/m9.figshare.26044609.v1 (accessed on 15 June 2024). 3. Data of 132 patients collected for predicting cardiac events: The dataset extending Table 5 is available on figshare at https://doi.org/10.6084/m9.figshare.26044594.v1 (accessed on 15 June 2024). 4. Comprehensive tools for ECG signal analysis: A GitHub repository containing tools for ECG signal analysis, including signal filtering, clustering, PQRST wave detection, and anomaly detection, as well as advanced machine learning algorithms for analyzing medical data, is available at https://github.com/zhadyralimbay/ECG-signal-analysis (accessed on 15 June 2024). These datasets and tools ensure that research results can be found, made available, compatible, and reused (FAIRLY) in accordance with the data exchange policy. Researchers are encouraged to use these resources for further analysis and research. Ethical and legal considerations have been taken into account, and the data does not compromise the anonymity of participants or violate local data protection laws.

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
