# Peer review of "Wearable ECG Device and Machine Learning for Heart Monitoring"

_sensors, 2024, doi:10.3390/s24134201_

Round 1
Reviewer 1 Report
Comments and Suggestions for Authors
Referee letter:
In this manuscript, Zhadyra Alimbayeva et al. proposed an innovative ECG monitoring system based on a single – lead ECG machine enhanced with machine learning methods. This device is designed to process and analyses the ECG data, but also predict potential heart disease at an early stage. This authors also construct an acquisition system and a server module with web-based segment of the system. Various machine learning methods was also conducted to predict cardiovascular disease. Some comments are listed below.
The main criticisms are:
Comment 1. The retrieved signals from wearable ECG device illustrated in figure 4 and figure 5 is not compatible with the clinical ECG. The low frequencies, including the P wave, U wave and P-R interval is not obvious. The R-peak is so small in the figure. And please annotate the Y-axis with unite of measure. If all ECG signal figure from the wearable ECG device is the same with the figure 4, I don’t think it is an effective ECG measure device. More figures from different diseases should be illustrated.
Comment 2. The parameter set for filtering the ECG signals may be wrong. As stated in Section 3.1, ‘the low cut range is 0.5Mhz’ and ‘the high cut range was set at the level of 50.0 MHz’. However, the ECG frequency in a normal person should in the range from 0.5Hz to 100Hz. The sampling frequency of ECG signal is 360 MHz, if it is real, the dataset may be too large. Please recheck your sampling frequency and other parameters in the filter section.
Comment 3. In Section 3.5, the figure 12 shows the correlation between each pair of features. The reader may be confused by the conclusion from the figure, where the coefficient between condition and others is less than 0.6. It cannot prove the conclusion that there are some relations between condition with other factors with a convincing proof.
Comment 4. The article utilizes the K-means to classify signal points. However, the recognition of PQRST wave should in the original sign, not in the segment of ECG signals by K-Means in clinical ECG. Please state the advantage, necessity and innovation in your method.
Comment 5. It is recommended for the authors to polish the language, especially pay attention to the superscript. In Section 2.4, the phase ‘the cluster μj’ should be expressed as ‘the cluster ’.
Comment 6. I didn't find anything innovative in your paper. It is hoped that you can compare your work with your counterpart to introduce your innovative points.
Comment 7. Refinement of language could be improved.
Comments on the Quality of English Language
None
Author Response
Thank you very much for taking the time to review this manuscript.
Please see the attachment

Reviewer 2 Report
Comments and Suggestions for Authors
This paper describes an ECG monitoring system including machine learning methods for detecting abnormalities. The system processes and analyses the ECG data, and also predict potential heart disease.
Although it addresses an interesting topic, the manuscript presents several flaws that should be fixed before publishing it.
1. The main problem with the article is that it is not clear what the innovative contributions of this work are over what currently exists in the literature and commercial devices.
It is also not clear if the different stages of sensorization, noise reduction and component extraction are innovations on what exists. If not, the explanation is too extensive and contributes little to the overall work.
2. In the introduction section the authors review several works on wearable ECG devices and artificial intelligence techniques applied to detect anomalies.
Although the intention seems to identify weak points and gaps that the authors will fill them with their proposal, at the end of the section it is not clear what these gaps are in the literature. In line 120, the authors state that "the review demonstrates the need to develop a wearable system..." but nothing can be concluded from the previous paragraphs.
The authors must emphasize what weaknesses currently exist and that they intend to address with their proposal.
3. On the other hand, the review is not exhaustive (it is not systematic) so we cannot conclude that the conclusions drawn by the authors are correct.
Interesting works may have been left unreviewed. In addition, works prior to 2020 are included that, although interesting, may not constitute the most current state of the art since in a field as revolutionary as AI, every year there are new, increasingly advanced proposals.
4. Section 2.2 would be improved with a Figure of components and involved technologies
5. Table 1 should include how many patients present each characteristic.
6. More explanation is needed for understanding tables 3 and 4.
7. The isolation forest method is applied and described the obtained results (20/56 records classified as anomalous), but how many records were anomalous? i.e., which are the accuracy, true positive rate and false positive rate?
8. What is the method used for calculate coorelation between demographic variables? After Figure 12, smoking is not referred as variable relevant for conditions (more than ECG change). Results from Figure 13 should be substituted by proper statistical analysis.
9. Dicussion and Conclusion sections repeat some ideas about the system, consider reduce and not repeat sentences already mentioned.
Comments on the Quality of English Language
The language and syntax of the entire article should be reviewed since it is difficult to understand. Additionally, there are many syntax and lexical errors.
Some minor comments (list not exhaustive):
- 22: leaning > learning
- 37, 39, 41: references supporting statements are missing
- 42: comparted > compared
- 54: 'The system...' > What system? In general every ECG system or the proposed in this paper? Please, rephrase it. Maybe some sentence is missing to make clear the content.
- 56 and others: arrhythmiac > arrhythmia (?)
- 84: 'appliances) low' > 'appliances) and low'
- 94: 'actively being studied' >'actively studied'; 'The paper' > 'The papers'
- 113: what 'CT' means?
- 133: wrong syntax
- 153: delete one 'ECG'
- 154: 'microcontrollers' > 'microcontroller'
Author Response

(The authors gave the same response as above.)

Reviewer 3 Report
Comments and Suggestions for Authors
This paper introduces a wearable single – lead ECG monitoring device that works in real time and can warn patients early about heart conditions. The device processes and analyses data, and then uses machine learning algorithms to predict heart disease at an early stage. The data are uploaded in real time to a web-based segment. The authors have used multiple algorithms to process the EGC data. For the prediction part, the authors provide a comparative analysis using various machine learning methods, with Convoluted Neural Networks (CNN) showing a prediction accuracy of 0.926.
The authors explain well how the proposed ECG monitoring system is an improvement over the state of the art. The authors state that this is the first ECG system in its kind, covering all the steps of monitoring process. The authors provide the block diagram of the device and describe the function of each part. The number of subjects tested is high; this gives confidence that the results are trustworthy.
Major comments
· Can the authors comment on the control experiment used to validate the predictions of the machine learning algorithms? Were conventional (manual) methods used, and if so which ones?
· Lines 210-250: It seems that a lot of the filters filter similar areas of the spectrum. For example, low-frequency components are filtered out both in the high-pass and in the band-pass filter. It is unclear to my why the authors use low-pass, high-pass and band-pass filters in the same circuit. Can the authors briefly explain e.g., why only using a band-pass filter is not enough to get the desired outcome?
· Line 379: Can the authors provide a brief explanation in the paper for why they chose five clusters? Is it because the peaks are five (PQRST)?
· Lines 307-310: Can the authors provide a brief explanation in the paper for how they chose the parameters for the isolation forest?
· Figure 11: What does the y-axis (index) represent? Is this the amplitude of the signal? The box graphs look similar between the different points (indicating similar statistics), even though e.g., R peaks have a much bigger amplitude than P, Q, S, T points. Can the authors comment on this?
Minor comments
· Line 113: “Moreover, a basic model with repetitive neural network (CT) and a lightweight model with cast CT was presented”. There is something wrong with the acronyms. Do the authors mean Recurrent Neural Network (RNN)?
· Line 223: “Antialiasing filters before sampling signals to prevent antialiasing” Anti-aliasing filters prevent aliasing, not anti-aliasing.
· Figure 1: “Anti-aliasing filter” instead of “Anti-Alice”
· Table 1: “diabetes” instead of “diabets”
· Figure 4: The names of the axes and the legends are too small to read. Also, although the legend shows two curves (blue, orange), the graph only shows the orange curve.
· Figure 6: The figure caption should briefly describe the logic behind the clustering and justify why they chose 5 clusters.
· It is my impression that Figure 14 and Table 6 convey the same information and can be combined. I would propose to keep the Figure and include in it the numbers from the table.
· The research data supporting this study should be uploaded to an online repository, such as Figshare or Zenodo, instead of a private Google Drive. This will also allow them to receive their own DOI. The code should be uploaded to a GitHub repository.
· Assuming the research is patented, can the authors provide the electronic schematics of their device in the SI for the purposes of reproducibility?
Comments on the Quality of English Language
There are several grammatical and/or spelling errors that need to be corrected during the proofs stage.
Author Response

(The authors gave the same response as above.)

Round 2
Reviewer 1 Report
Comments and Suggestions for Authors
After the revision, the manuscript can be accepted.
Comments on the Quality of English Language
None
Author Response
Thank you very much for taking the time to review this manuscript. Please find the detailed responses below and the corresponding revisions in the re-submitted files.
Comments 1: After the revision, the manuscript can be accepted.
Response 1: We utilized the editing services of MDPI for a thorough revision of the manuscript in English. The certificate of editing and the revised manuscript are attached.

Reviewer 2 Report
Comments and Suggestions for Authors
The authors' contributions to the topic are now better understood.
One of them is the possibility for the patient to use the system freely, but would the assistance of a professional be necessary to place the electrodes? Could a patient use the device autonomously on a daily basis and know how to place the electrodes in their correct position? Have you analyzed the usability of the device?
The statements before and after table 3 (and those before and after table 4) are repetitive.
Appendix A may be too detailed for this work.
Does Appendix B show personal data? They should be omitted or, where appropriate, indicate that they are fictitious data.
Comments on the Quality of English Language
No comments.
Author Response
Thank you very much for taking the time to review this manuscript. Please find the detailed responses below and the corresponding revisions in the re-submitted files
Comments 1: One of them is the possibility for the patient to use the system freely, but would the assistance of a professional be necessary to place the electrodes? Could a patient use the device autonomously on a daily basis and know how to place the electrodes in their correct position? Have you analyzed the usability of the device?
Response 1: We have developed a carrying case for the ECG device and a strap for its convenient wearing, which significantly simplifies the daily use of the device. The device is accompanied by a detailed user manual, allowing patients to independently place the electrodes without professional assistance. In our research and practical experience, we have confirmed that patients can independently place the electrodes using the provided instructions. Therefore, professional assistance is not required for this task. However, we acknowledge that there is an issue with skin irritation at the electrode attachment sites after 48 hours of wearing the device. This is related to the use of disposable electrodes made from foam material with an Ag/AgCl sensor. To reduce the risk of skin irritation, we strive to use electrodes coated with hypoallergenic adhesive. Despite this, the issue persists. Therefore, we continue to research and test electrodes from different manufacturers to find the optimal solution. Additionally, we are working on developing our own electrodes that will be more comfortable and safer for patients
Comments 2: The statements before and after table 3 (and those before and after table 4) are repetitive.
Response 2: Thank you for your comments. The redundant statements have been removed
Comments 3: Appendix A may be too detailed for this work. Does Appendix B show personal data? They should be omitted or, where appropriate, indicate that they are fictitious data.
Response 3: Thank you for your comments. We included a detailed electrical schematic in Appendix A and the user interface of the ECG monitoring system software in Appendix B to validate the accuracy of our results. However, if you do not object, we would like to submit the manuscript without these details to preserve commercial confidentiality and protect personal data. In the revised manuscript, the appendices have been removed.
We also utilized the editing services of MDPI for a thorough revision of the manuscript in English. The certificate of editing and the revised manuscript are attached.
